# Myeloid-derived miR-6236 potentiates adipocyte insulin signaling and prevents hyperglycemia during obesity

Bam D. Paneru[1,11], Julia Chini [1,2,11], Sam J. McCright [1,2], Nicole DeMarco[1], Jessica Miller[1], Leonel D. Joannas[3], Jorge Henao-Mejia [3], Paul M. Titchenell [4,5], David M. Merrick[6], Hee-Woong Lim [7,8], Mitchell A. Lazar [5,6] & David A. Hill [1,5,9,10] ✉

Adipose tissue macrophages (ATMs) influence obesity-associated metabolic dysfunction, but the mechanisms by which they do so are not well understood. We show that miR-6236 is a bona fide miRNA that is secreted by ATMs during obesity. Global or myeloid cell-specific deletion of miR-6236 aggravates obesity-associated adipose tissue insulin resistance, hyperglycemia, hyper-insulinemia, and hyperlipidemia. miR-6236 augments adipocyte insulin sensitivity by inhibiting translation of negative regulators of insulin signaling, including PTEN. The human genome harbors a miR-6236 homolog that is highly expressed in the serum and adipose tissue of obese people. hsa-MIR-6236 expression negatively correlates with hyperglycemia and glucose intolerance, and positively correlates with insulin sensitivity. Together, our findings establish miR-6236 as an ATM-secreted miRNA that potentiates adipocyte insulin signaling and protects against metabolic dysfunction during obesity.

Adipose tissue has evolved to store energy during periods of excess caloric intake and release energy during periods of caloric deprivation. This process is tightly regulated by insulin, a hormone that promotes lipogenesis and inhibits lipolysis with the ultimate function of maintaining global energy homeostasis[1]. However, periods of extended and excessive caloric intake can lead to obesity, tissue insulin resistance, and the development of type 2 diabetes mellitus (T2DM)[2]. Though the mechanisms of this pathologic cascade are not completely understood, adipose tissue insulin resistance may directly contribute to T2DM[3]. As a result, molecules that restore adipose tissue insulin sensitivity are of high interest as potential T2DM therapeutics[4].

Adipose tissue undergoes dramatic remodeling during obesity including adipocyte hypertrophy, increased vascularization, and immune cell accumulation[5]. The past decade has witnessed a revolution in our understanding of how immune cells influence adipose tissue remodeling and function in the context of obesity[6]. Adipose tissue macrophages (ATMs) are now understood to be a heterogeneous immune cell population that can have harmful or beneficial effects on adipose tissue function depending on context[7]. For example, we recently identified lipid-associated macrophages (LAMs), a metabolically-activated ATM population that accumulates in obese adipose tissue of mice and humans[8,9]. This ATM subset promotes

[1]Division of Allergy and Immunology, Children's Hospital of Philadelphia, Philadelphia, PA, USA. [2]Medical Scientist Training Program, University of Pennsylvania Perelman School of Medicine, Philadelphia, PA, USA. [3]Department of Pathology and Laboratory Medicine, University of Pennsylvania Perelman School of Medicine, Philadelphia, PA, USA. [4]Department of Physiology, University of Pennsylvania Perelman School of Medicine, Philadelphia, PA, USA. [5]Institute for Diabetes, Obesity and Metabolism, University of Pennsylvania Perelman School of Medicine, Philadelphia, PA, USA. [6]Department of Medicine, Division of Endocrinology, University of Pennsylvania Perelman School of Medicine, Philadelphia, PA, USA. [7]Division of Biomedical Informatics, Cincinnati Children's Hospital Medical Center, Cincinnati, OH, USA. [8]Department of Pediatrics, University of Cincinnati College of Medicine, Cincinnati, OH, USA. [9]Institute for Immunology and Immune Health, University of Pennsylvania Perelman School of Medicine, Philadelphia, PA, USA. [10]Department of Pediatrics, University of Pennsylvania Perelman School of Medicine, Philadelphia, PA, USA. [11]These authors contributed equally: Bam D. Paneru, Julia Chini. ✉e-mail: hilld3@chop.edu

adipose tissue function and global metabolic homeostasis, and is predominantly beneficial in the context of obesity[9]. However, the mechanisms by which ATMs exert their effects in adipose tissue are not well understood.

LAMs are characterized by surface expression of CD9[8], a tetraspanin involved in extracellular vesicle (EV) biogenesis, packaging, release, and uptake[10]. We have found that LAMs secrete large quantities of EVs[8]—a cellular feature central to ATM functions in other contexts[11,12]. One mechanism by which EV release may mediate ATM function is through the secretion and delivery of microRNAs (miRNAs) to neighboring cells. miRNAs are small, noncoding RNA molecules that are transcribed as primary miRNAs (pri-miRNAs), and subsequently cleaved to generate precursor miRNAs (pre-miRNAs) and mature miRNAs[13]. miRNAs regulate the translation of a target gene by binding to the 3′ untranslated region (UTR) of the mRNA product, leading to its suppression or degradation[14].

Given the diverse functions of ATMs during obesity and their ability to secrete EVs, we sought to investigate whether ATM-derived EVs contain metabolically relevant miRNAs. To do so, we performed small RNA sequencing (RNA-seq) of LAM-derived EVs and characterized a novel, ATM-secreted miRNA: miR-6236. We studied the molecular, cellular, and organismal functions of miR-6236 using novel loss-of-function mouse strains complemented by in vitro and in vivo gain-of-function approaches. We identified and validated the mRNA of

PTEN, a known negative regulator of insulin signaling, as a miR-6236 target. Finally, we analyzed human transcriptome and epigenome sequencing data to identify a homolog of miR-6236 and correlated its expression with key obesity-associated clinical outcomes. By doing so, we have characterized a mechanism by which ATMs regulate adipose tissue function and global metabolic homeostasis.

## Results

### miR-6236 is an ATM-secreted miRNA that is enriched in adipose tissue and increased during obesity

To determine the LAM miRNA secretome, we performed small RNA-sequencing (RNA-seq) on EVs purified from ex vivo LAM cultures isolated from wild-type (WT) mice subjected to a model of diet-induced obesity (DIO) (Supplementary Fig. 1a). We found that miR-6236 is the most abundant miRNA in LAM-derived EVs (Fig. 1a, b). While miR-6236 has been predicted computationally, its sequence, structure, and functions have not been previously validated. To validate miR-6236, we combined our sequencing data with legacy data from 324 previously published small RNA-seq libraries encompassing 19 tissues, ATMs, and ATM-derived EVs[11,15,16], and aligned reads from these libraries to the pre-miR-6236 locus. The majority of RNA-seq reads aligned to a region upstream of the predicted mature miR-6236 sequence (Fig. 1c). Next, we used legacy data to map adipose tissue Argonaute high-throughput sequencing of RNA isolated by

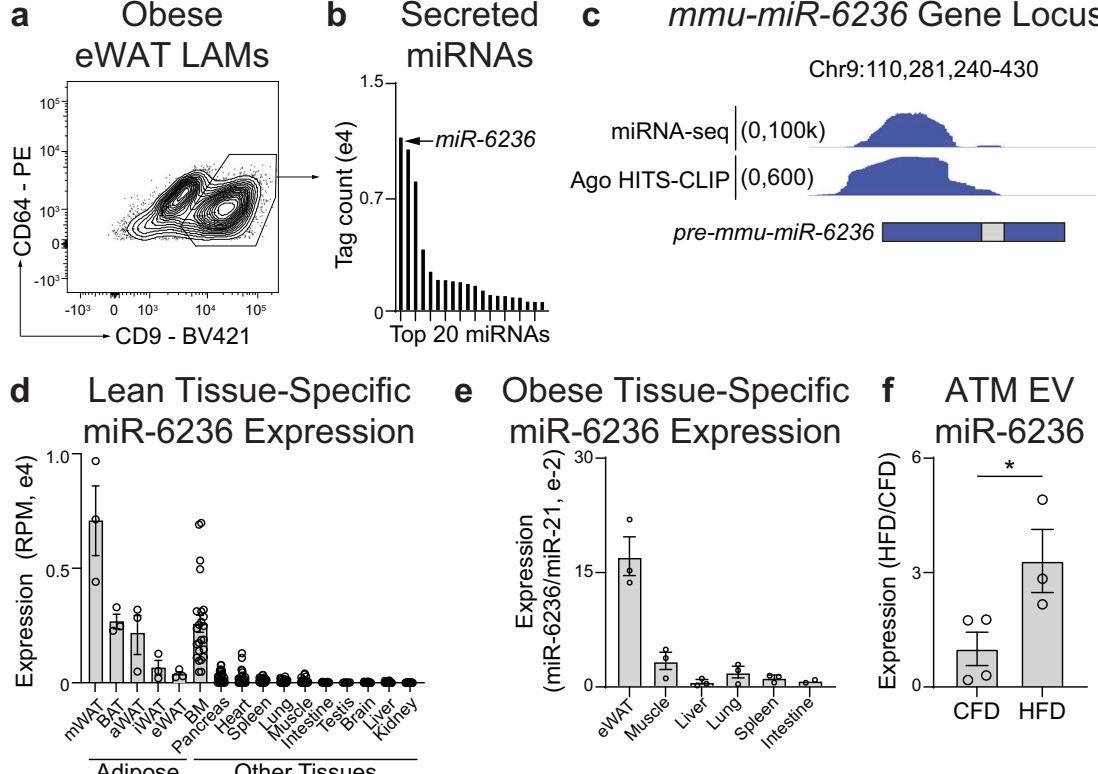

**Fig. 1 | miR-6236 is a LAM-secreted miRNA that is enriched in adipose tissue and regulated by obesity. a** Flow cytometric identification of CD9+ LAMs in epididymal white adipose tissue (eWAT) of obese mice. **b** The 20 most abundant miRNAs in the EV fraction of sort-purified CD9+ LAMs as detected by small RNA sequencing (RNA-seq). **c** Genomic alignment of reads from small RNA-seq and Argonaute HITS-CLIP. miR-6236 locus with historically annotated pre-miR-6236 and mature miR-6236 regions indicated via the blue and gray bars, respectively. **d** Expression level of miR-6236 in lean mouse adipose tissue depots (n = 3 biological replicates/tissue), BM (n = 24), Pancreas (n = 26), Heart (n = 26), Spleen (n = 28), Lung (n = 26), Muscle (n = 28), Intestine (n = 24), Testis (n = 6), Brain (n = 28), Liver (n = 28), and Kidney (n = 28) measured by small RNA-seq. **e** Expression level of miR-6236 in obese mouse

tissue measured by qPCR (n = 3 biological replicates/tissue, except intestine where n = 2). **f** Expression level of miR-6236 in EVs isolated from adipose tissue macrophages (ATMs) of lean (CFD; n = 4) or obese (HFD; n = 3) mice by small RNA-seq (p = 0.045). LAM lipid associated macrophage, Ago Argonaute, HITS-CLIP high throughput sequencing-crosslinking immunoprecipitation, RPM reads per million, mWAT mesenteric white adipose tissue, BAT brown adipose tissue, aWAT intra-abdominal white adipose tissue, iWAT inguinal white adipose tissue, eWAT epididymal white adipose tissue, BM bone marrow, CFD control fat diet, HFD high fat diet. Data representative of ≥2 experiments. Data in **d–f** presented as mean ± SEM; *p < 0.05, two-tailed Student's t test.

crosslinking immunoprecipitation (HITS-CLIP) reads to the *pre-miR-6236* locus[16]. Consistent with our RNA-seq alignment, HITS-CLIP reads predominantly mapped to a region upstream of the predicted mature miR-6236 sequence (Fig. 1c). Together, these data indicate that prior miR-6236 annotations are inaccurate and provide enhanced resolution of the *miR-6236* locus.

We next reannotated the *pre-miR-6236* sequence and found that *pre-miR-6236* has more than one predicted stem and loop, and several predicted mature sequence isoforms (Supplementary Fig. 1b, c). To determine the expression pattern of miR-6236, we reanalyzed legacy small RNA-seq data of lean WT mice[15,16]. This analysis revealed that miR-6236 expression was largely restricted to adipose tissue and bone marrow (Fig. 1d). A similar expression pattern was observed in the adipose tissue of DIO WT mice (Fig. 1e). To determine if miR-6236 expression is altered by obesity, we analyzed legacy small RNA-seq data from ATM-derived EVs[11]. We observed that miR-6236 was elevated in ATM-derived EVs of DIO WT mice compared to those of lean WT mice (Fig. 1f), indicating that miR-6236 expression is dynamically regulated by obesity.

As miR-6236 is a novel miRNA, we further characterized its expression pattern across various cell types and under different physiological conditions using newly generated and publicly available legacy data sets. First, we examined expression of miR-6236 in sort-purified epididymal white adipose tissue (eWAT) myeloid immune cells of DIO WT mice. We observed that miR-6236 was present in macrophages, monocytes, and neutrophils, but was expressed at a lower level in neutrophils compared with macrophages and monocytes (Supplementary Fig. 1d). To determine whether miR-6236 was secreted from macrophages and monocytes, we compared miR-6236 levels in EVs purified from cultures of DIO WT eWAT Ly6c+ monocytes, CD9- ATMs, and CD9+ LAMs[8]. We observed that miR-6236 was present in EVs derived from all myeloid subsets, but was enriched in EVs derived from ATMs and LAMs relative to monocytes (Supplementary Fig. 1e). As miR-6236 expression was enriched in EVs derived from ATMs and LAMs, we next examined the relative capacity for these two macrophage populations to secrete EVs. We found that LAMs secreted nearly twice as many EVs per cell as compared with CD9- ATMs (Supplementary Fig. 1f). As LAMs are characterized by CD9 expression, we also confirmed that LAM-derived EVs contain CD9 (Supplementary Fig. 1g). Finally, we examined the effect of age on miR-6236 expression by analyzing legacy small RNA-seq data from the serum of young (6 months) or aged (24 months) mice[17]. Consistent with secretion in EVs, miR-6236 was detected in the serum of these mice, but its levels were not significantly different between young and aged mice (Supplementary Fig. 1h).

To determine potential upstream regulators of miR-6236 expression, we next treated BMDMs with various cytokines and fatty acids and measured miR-6236 expression by qPCR. Fatty acids palmitate and arachidonate upregulated, while M1 (LPS + IFNγ) and M2 (IL-4) polarizing agents downregulated, the expression of miR-6236 (Supplementary Fig. 1i). Because we found that fatty acids stimulated miR-6236 expression, and the lipid receptor *Trem2* has been found to be a feature of LAMs[9], we tested if TREM2 influenced miR-6236 expression. TREM2 appeared to negatively regulate miR-6236 expression in naïve BMDMs but had no effect on miR-6236 expression in arachidonate-stimulated BMDMs (Supplementary Fig. 1j). To further explore the role of TREM2, and test if miR-6236 was expressed by other metabolic tissue macrophages, we analyzed legacy small RNA-seq data of Kupffer cells from DIO WT and *Trem2* knockout mice[18]. We detected miR-6236 in Kupffer cells (Supplementary Fig. 1k), and Kupffer cell miR-6236 expression did not require the presence the of *Trem2*. Finally, we examined potential epigenetic mechanisms of regulation of the *pre-miR-6236* locus using legacy data to map macrophage PPARγ chromatin immunoprecipitation with sequencing (ChIP-seq) and LAM assay for transposase-accessible chromatin with sequencing (ATAC-

seq) reads to the *pre-miR-6236* locus[8,19]. We found overlapping ATAC-seq and PPARγ ChIP-seq peaks at the *pre-miR-6236* locus (Supplementary Fig. 1l), suggesting a potential means of transcriptional regulation by PPARγ. Consistently, rosiglitazone, a PPARγ agonist, induced miR-6236 expression in arachidonate-stimulated BMDMs (Supplementary Fig. 1m). Together, these data indicate that miR-6236 is a novel, myeloid-secreted miRNA that is enriched in adipose tissue and positively regulated by obesity and metabolic macrophage activation.

## miR-6236 protects against the development of obesity-associated metabolic outcomes

To investigate the in vivo functions of miR-6236, we generated a whole-body miR-6236 knockout (KO) mouse line (Supplementary Fig. 2a), and evaluated obesity-associated outcomes by subjecting KO and WT littermate control mice to a DIO model[8]. We found that the *miR-6236* locus and miR-6236 molecule were deleted in KO mice as indicated by DNA gel genotyping and miR-6236 qPCR of BMDM-derived EVs (Supplementary Fig. 2b, c). Though body weight did not differ between DIO WT and DIO KO male mice (Fig. 2a), DIO KO male mice had elevated fed and fasting serum glucose levels, elevated fed serum insulin levels, and worse fasting glucose tolerance compared to obese WT male mice (Fig. 2b–d). DIO KO male mice also had elevated serum free fatty acid and glycerol levels compared with DIO WT male mice (Fig. 2e, f). A similar phenotype was observed in DIO female KO mice (Supplementary Fig. 2d–f). Body weight, blood glucose, serum insulin, and glycerol levels were not different between WT and KO male mice fed normal chow diet for 18 weeks (Supplementary Fig. 2g–j), and serum glucagon, triglyceride, and cholesterol levels did not differ between DIO WT and DIO KO mice (Supplementary Fig. 2k–m). These findings suggest that miR-6236 is acting to support insulin sensitivity and glucose homeostasis during DIO.

To further interrogate the metabolic derangements observed in KO mice, we used a hyperinsulinemic-euglycemic clamp to measure insulin sensitivity. These studies were performed with +/− or −/− littermates that were co-housed in mixed cages, to minimize variability due to environmental factors. Briefly, mice received a constant infusion of insulin and radiolabeled glucose was co-infused, as needed, to maintain euglycemia. DIO KO mice required a lower glucose infusion rate to achieve euglycemia (Fig. 2g, Supplementary Fig. 2n), and had a non-significant trend towards lower peripheral glucose disposal (Supplementary Fig. 2o), as compared with DIO littermate controls. Further, measurement of radiolabeled glucose accumulation revealed a non-significant trend towards a defect in glucose uptake by eWAT, but not other adipose tissue depots, in DIO KO mice as compared to DIO littermate controls (Supplementary Fig. 2p). We also observed a non-significant trend towards a defect in glucose uptake by vastus lateralis, but not other muscle depots, of DIO KO mice as compared to DIO littermate controls (Supplementary Fig. 2q). We did not observe a difference in basal or insulin-mediated suppression of hepatic glucose production between DIO KO and littermate control mice (Supplementary Fig. 2r).

## miR-6236 promotes insulin-mediated signaling and functions in adipocytes

Given the global metabolic changes observed in DIO KO mice, we next examined the cellular, molecular, and metabolic phenotype of key metabolic tissues in DIO WT and DIO KO mice. The weight of eWAT and liver did not differ between DIO WT or DIO KO mice (Supplementary Fig. 3a), though livers of DIO KO mice had higher triglyceride content as compared with those of DIO WT mice (Supplementary Fig. 3b). Further, both liver and eWAT of DIO KO mice had non-significant decreases in the expression of de novo lipogenesis genes as compared with those of DIO WT mice (Supplementary Fig. 3c). Upon examination of adipose tissue histology, we observed that eWAT

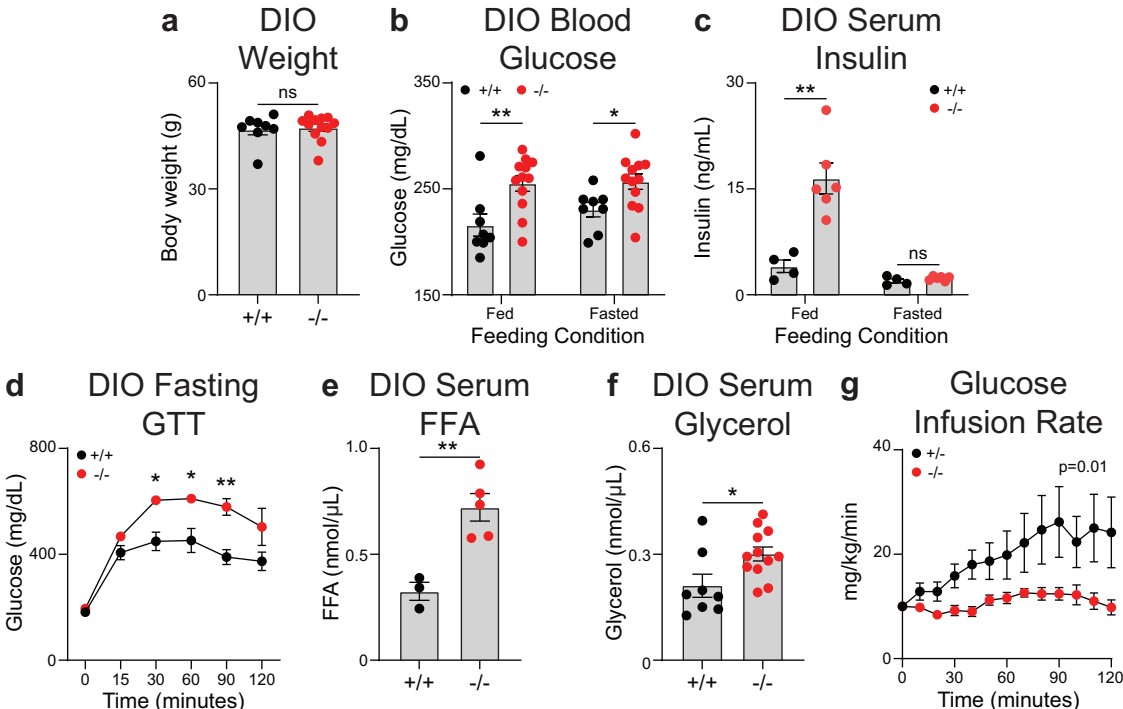

**Fig. 2 | Whole body deletion of miR-6236 exacerbates obesity-associated metabolic outcomes. a** Body weight of wild-type (+/+; *n* = 8) or miR-6236 knockout (−/−; *n* = 12) male mice subjected to diet-induced obesity (DIO). **b** Fed and fasted (5 h) blood glucose level in DIO male wild-type (*n* = 8) and knockout (*n* = 12) mice (Fed, *p* = 0.0055; Fasted, *p* = 0.022). **c** Fed and fasted (5 h) serum insulin level in DIO male wild-type (*n* = 4) and knockout (*n* = 6) mice (Fed, *p* = 0.0023). **d** Intraperitoneal glucose tolerance test (GTT) in DIO male wild-type (*n* = 6) and knockout (*n* = 3) mice (30 min., *p* = 0.018; 60 min., *p* = 0.049; 90 min., *p* = 0.0047).

**e** Serum free fatty acid (FFA) level in DIO male wild-type (*n* = 3) and knockout (*n* = 5) mice (*p* = 0.0051). **f** Serum glycerol level in DIO male wild-type (*n* = 8) and knockout (*n* = 12) mice (*p* = 0.022). **g** Glucose infusion rate during a fasting hyperinsulinemic-euglycemic clamp in DIO male mice of +/− (*n* = 6) or −/− genotypes (*n* = 5) (*p* = 0.011). Data representative of ≥2 experiments with exception of **g** which is 1 experiment. Data presented as mean ± SEM; ns not significant, \**p* < 0.05, \*\**p* < 0.01, two-tailed Student's *t* test (**a**–**f**) and two-tailed two-way ANOVA with interaction term (**g**).

adipocyte size was slightly larger in DIO KO as compared with DIO WT mice (Supplementary Fig. 3d), while inguinal white adipose tissue (iWAT) adipocyte size was unchanged (Supplementary Fig. 3e). Together, these data indicate that DIO KO mice display some features of exaggerated tissue insulin resistance compared with DIO WT mice.

In adipocytes, insulin has multiple functions that maintain metabolic homeostasis including promoting glucose uptake and lipogenesis, and inhibiting lipolysis[1]. Given our prior findings, we hypothesized that miR-6236 was influencing adipocyte insulin signaling. Expression of the insulin receptor (INSR) in eWAT did not differ between DIO WT and DIO KO mice (Supplementary Fig. 3f). However, phosphorylation of AKT serine/threonine kinase 2 (pAKT2) was significantly lower in eWAT of DIO KO as compared with DIO WT mice (Fig. 3a). To better understand the functional consequences of this defect, we examined readouts of insulin signaling in adipocytes including glucose uptake and lipolysis. Ex vivo insulin-stimulated glucose uptake was lower in adipocytes isolated from eWAT of DIO KO as compared with DIO WT mice (Fig. 3b), while lipolysis was increased (Fig. 3c). Consistently, expression of genes central to lipolysis were elevated in eWAT of DIO KO as compared to DIO WT mice (Fig. 3d), as were levels of phosphorylated hormone-sensitive lipase (pHSL) (Fig. 3e). Notably, pAKT2 phosphorylation was not altered in eWAT of lean KO mice as compared to lean WT mice (Supplementary Fig. 3g), an observation consistent with observed blood glucose and insulin measurements (Supplementary Fig 2h, i). In sum, these findings are consistent with reduced adipocyte insulin signaling in DIO KO as compared with DIO WT mice.

To further establish the effects of miR-6236 on insulin-dependent adipocyte functions, we performed gain-of-function studies by transfecting miR-6236 mimic or control miRNA into in vitro differentiated 3T3-L1 adipocytes. Transfection with miR-6236 mimic increased 3T3-L1

adipocyte AKT phosphorylation, insulin-stimulated glucose uptake, and insulin-mediated suppression of lipolysis (Supplementary Fig. 3h–j). To determine if macrophage-derived EVs were sufficient to mediate the effects of miR-6236 on adipocyte insulin-dependent functions, we metabolically-activated bone marrow-derived macrophages (BMDMs) derived from KO or WT mice with palmitate, purified BMDM-secreted EVs, and cultured the EVs with differentiated WT primary murine adipocytes. Treatment of primary adipocytes with EVs isolated from WT BMDMs resulted in increased AKT phosphorylation, increased insulin-stimulated glucose uptake, and enhanced insulin-mediated suppression of lipolysis, as compared to treatment with EVs isolated from KO BMDMs (Fig. 3f–h). Finally, to investigate the effects of EV-delivered miR-6236 in vivo, we intraperitoneally (i.p.) injected WT or KO BMDM-derived EVs into DIO WT mice and measured insulin-dependent phenotypes including serum glucose levels, serum insulin levels, and eWAT pAKT2 levels. Consistent with our hypothesis that WT EVs (that contain miR-6236) would potentiate insulin signaling upon transfer into obese mice, mice that received WT EVs had reduced blood glucose and serum insulin levels as compared to mice that received KO EVs (Fig. 3i, j). However, we did not detect differences in eWAT AKT phosphorylation in mice treated with WT versus KO EVs (Supplementary Fig. 3k). Together, these results indicate that macrophage-derived EVs that lack miR-6236 have an impaired ability to promote insulin-mediated signaling and global glucose homeostasis.

### miR-6236 augments insulin signaling in adipocytes by inhibiting PTEN expression
We next sought to determine a mechanism by which miR-6236 regulates insulin signaling in adipocytes. miRNAs bind to the 3′UTR of

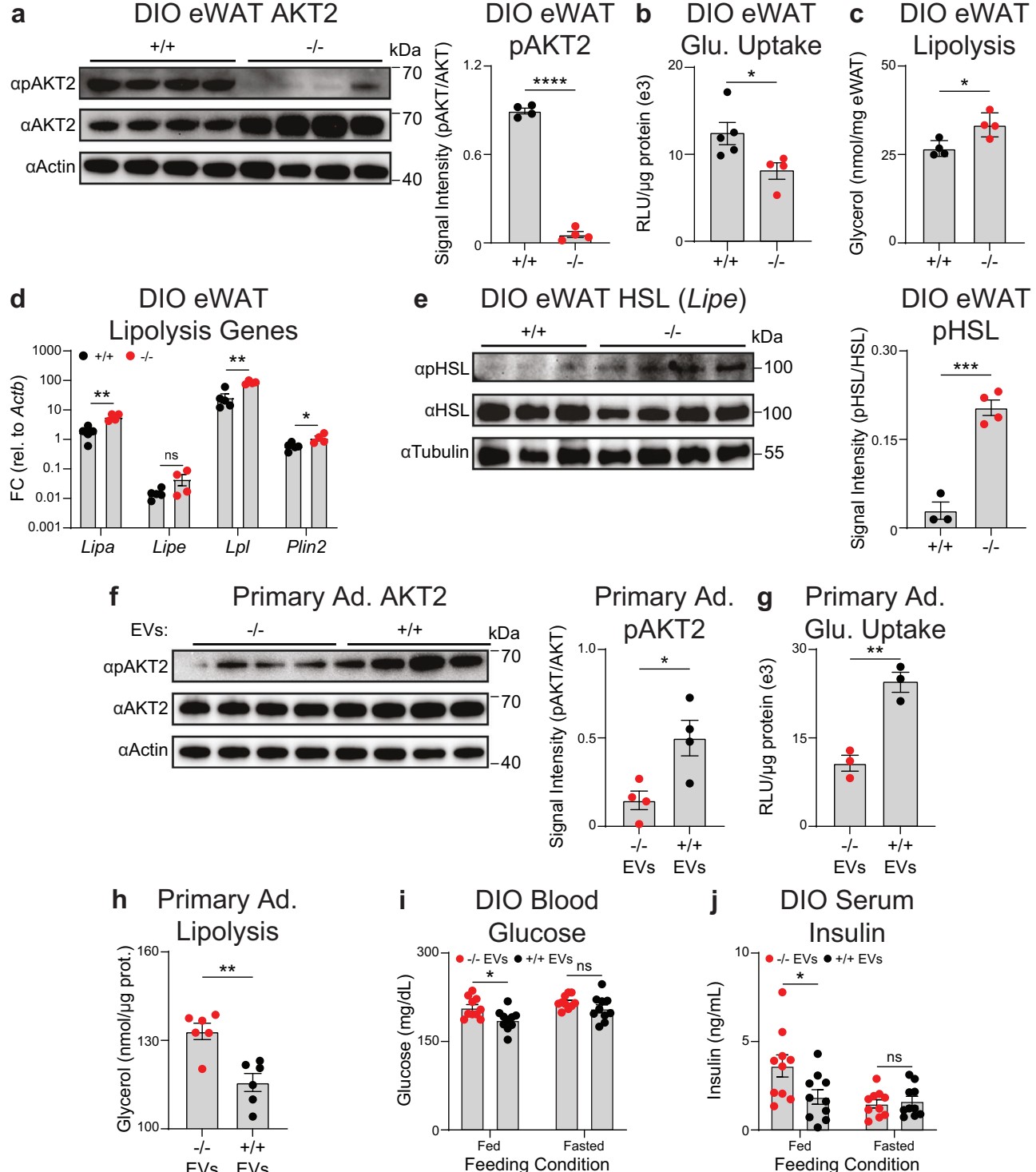

**Fig. 3 | miR-6236 promotes insulin-mediated signaling and functions in adipocytes. a** AKT2 and pAKT2 Western blot and quantification in eWAT of DIO wild-type (+/+) or miR-6236 knockout (−/−) male mice ($n = 4$; $p < 0.0001$). **b** Insulin-stimulated glucose (Glu.) uptake in mature adipocytes isolated from eWAT of DIO +/+ ($n = 5$) or −/− ($n = 4$) male mice ($p = 0.037$). RLU Relative light units. **c** Glycerol release from eWAT adipocytes of DIO +/+ or −/− male mice in the presence of insulin ($n = 4$; $p = 0.017$). **d** Expression of key lipolysis genes in eWAT of DIO +/+ ($n = 5$) or −/− ($n = 4$) male mice (*Lipa*, $p = 0.0016$; *Lipe*, $p = 0.1044$; *Lpl*, $p = 0.0011$; *Plin2*, $p = 0.032$). **e** HSL and pHSL Western blot and quantification in eWAT of DIO +/+ ($n = 3$) or −/− ($n = 4$) male mice ($p = 0.0003$). **f** AKT2 and pAKT2 Western blot and quantification in +/+ primary adipocytes (Ad.) treated with EVs isolated from −/− or +/+ BMDMs ($n = 4$ technical replicates, $p = 0.021$). **g** Insulin-stimulated glucose uptake in +/+ primary adipocytes treated with EVs isolated from −/− or +/+ BMDMs ($n = 3$ technical replicates, $p = 0.0033$). **h** Glycerol release from +/+ primary adipocytes treated with EVs isolated from −/− or +/+ BMDMs ($n = 6$ technical replicates, $p = 0.0018$). **i–j** Fed and fasted (5 h) blood glucose (**i**) and serum insulin (**j**) levels ($n = 10$ mice/group, Fed Glucose, $p = 0.016$; Fed Insulin, $p = 0.030$) in DIO +/+ male mice injected i.p. with EVs from −/− or +/+ BMDMs. Data representative of ≥2 experiments. Data presented as mean ± SEM; ns not significant, *$p < 0.05$, **$p < 0.01$, ***$p < 0.001$, ****$p < 0.0001$, two-tailed Student's $t$ test.

target mRNAs leading to suppression of translation or mRNA degradation[14]. To identify putative miR-6236 targets, we performed computational assessment of predicted miR-6236 binding sites in the 3′UTRs of mRNA transcripts genome-wide using the sRNAtoolbox package[20]. We then focused our validation efforts on targets known to negatively regulate insulin signaling (Supplementary Fig. 4a). One such potential target is *Pten*, the mRNA product of which contains 7 predicted miR-6236 binding sites in its 3′UTR. PTEN antagonizes PI3K (Phosphoinositide 3-kinase) by dephosphorylating phosphatidylinositol-3,4,5-triphosphate, thereby inhibiting insulin signaling[21,22]. Various knockout animal studies have shown that interfering with PTEN action improves insulin-dependent cellular metabolism, global insulin sensitivity, and glucose homeostasis[23]. As such, PTEN was deemed to have strong potential to be post-transcriptionally regulated by miR-6236 and therefore, contribute to the phenotype observed in miR-6236 KO mice.

To determine if PTEN is post-transcriptionally regulated by miR-6236, we first sought to determine if miR-6236 binds to the 3′UTR of the *Pten* mRNA. To do so, we cloned a ~1 kb region of the *Pten* 3′UTR harboring 2 of the 7 predicted miR-6236 binding sites downstream of the firefly luciferase gene open reading frame (WT 3′UTR) (Supplementary Fig. 4b, c). We also generated a second construct that lacked the two predicted binding sites in the *Pten* 3′UTR (Mut 3′UTR). We then tested the effects of a miR-6236 mimic or a control siRNA on luciferase expression in vitro. We found that the miR-6236 mimic reduced firefly luciferase expression from the WT *Pten* 3′UTR construct, but not from the Mut *Pten* 3′UTR construct, indicating that miR-6236 bound to the predicted target sites in the 3′UTR of the *Pten* mRNA and interfered with downstream gene translation (Fig. 4a).

We next sought to determine if the miR-6236 mimic influences PTEN protein levels and insulin signaling in adipocytes. To do so, we transfected differentiated 3T3-L1 adipocytes with control siRNA or miR-6236 mimic and measured PTEN protein levels. Upon transfection of 3T3-L1 adipocytes with the miR-6236 mimic, we observed a reduction in PTEN protein levels (Fig. 4b) with an appropriate compensatory increase in AKT phosphorylation (Supplementary Fig. 4d). The effects of miR-6236 were similar to those observed with a small interfering RNA targeting the *Pten* mRNA (siPten), and there were no additive effects of miR-6236 and siPten (Fig. 4b and Supplementary Fig. 4d), suggesting that miR-6236 and the *Pten* siRNA acted through the same mechanism to influence AKT phosphorylation. Furthermore, miR-6236 and siPten transfection resulted in similar increases in 3T3-L1 adipocyte glucose uptake and lipid accumulation without additive effects (Fig. 4c, d), further supporting that miR-6236 acts through PTEN to influence adipocyte functions. We next sought to test the effects of miR-6236 on adipose tissue PTEN levels in vivo. PTEN protein levels were elevated in eWAT of DIO KO as compared with DIO WT mice (Fig. 4e). In contrast, there was a non-significant trend towards reduced PTEN protein levels in eWAT of DIO WT mice that received i.p. injections of WT EVs, compared to mice that received miR-6236 deficient EVs (Fig. 4f). Together, these data validate PTEN as being post-transcriptionally regulated by miR-6236 in a manner that can augment insulin signaling in adipocytes and adipose tissue.

As miR-6236 is predicted to bind multiple targets (Supplementary Fig. 4a)[24], the function of miR-6236 may not be mediated solely via suppression of PTEN. As such, we also experimentally validated another top-predicted target of miR-6236, *Prkca*. We observed that miR-6236 binds to the 3′UTR of *Prkca* and suppresses subsequent gene translation (Supplementary Fig. 4e). As such, miR-6236 may regulate other molecules relevant to adipocyte insulin signaling during obesity.

## miR-6236 does not have major effects on immune cell populations in peripheral compartments

Given the high expression of miR-6236 in macrophages and bone marrow, we investigated how loss of miR-6236 affects the frequency and distribution of various immune cells in major immunological organs including the bone marrow (BM), blood, and spleen. The gating strategy used to profile different immune cell progenitors and mature myeloid and lymphoid lineages is provided in Supplementary Figs. 5, 6. The number of CD45+ cells was reduced in BM of DIO KO compared to DIO WT mice (Supplementary Fig. 7a). Among immune cell progenitors, the frequency and number of common myeloid progenitors (CMP) were slightly reduced in KO mice, while the frequency and number of the other immune cell progenitors were unchanged (Supplementary Fig. 7b). Among mature bone marrow immune cells, the frequency and number of monocytes and neutrophils were reduced in KOs, though monocyte frequency as a percentage of total CD45+ cells was not significantly different between WT and KO mice (Supplementary Fig. 7c). In the blood and spleen, neither the total number of CD45+ cells nor the percentage or number of any specific mature immune cell population was altered between DIO WT and DIO KO mice (Supplementary Fig. 7d–g). Together, these data indicate that miR-6236 has minimal effects on the development and distribution of immune cells in peripheral organs.

## Myeloid cell-derived miR-6236 influences adipose tissue insulin signaling and global metabolic homeostasis during obesity

As miR-6236 is secreted by ATMs, we hypothesized that ATM-derived miR-6236 regulates adipocyte insulin signaling during obesity in an ATM-extrinsic manner. However, it is possible that miR-6236 has cell-intrinsic functions in ATMs and/or adipocytes that influence adipocyte insulin signaling[8,9]. To preliminarily test for immune cell-intrinsic functions of miR-6236, we profiled eWAT immune cell populations in DIO WT or DIO KO mice. We observed that adaptive and innate immune cell populations, including LAMs, were present in similar numbers in DIO WT and DIO KO mice (Supplementary Figs. 5a, b, 8a). However, LAMs of DIO KO mice displayed higher levels of intracellular lipid (Supplementary Fig. 8b), consistent with their known functions related to lipid phagocytosis and the increased lipolysis observed in DIO KO mice[7,8]. To test if miR-6236 influences macrophage activation or functional states, we next carried out ex vivo M1, M2, or metabolic activation of WT or KO BMDMs. Expression levels of key M1 (*Stat1* and *Tnfa*), M2 (*Arg1* and *Cd206*), and metabolic activation (*Cd9* and *Lipa*) transcriptional markers were unaltered between appropriately polarized WT or KO BMDMs (Supplementary Fig. 8c–e). Given that PTEN is a known tumor suppressor, we also investigated if miR-6236 affects cell proliferation. Upon culture of BMDMs ex vivo, we did not observe a difference in the capacity for cell proliferation between WT and KO BMDMs (Supplementary Fig. 8f). To test if miR-6236 influences ATM polarization in vivo, we next investigated expression of select macrophage-related genes in sort-purified ATMs or LAMs from DIO WT or DIO KO mice. We observed minimal changes in expression of key ATM-related genes (*Trem2, Lipa, Tnf, Tlr2,* and *Arg1*) between DIO WT and DIO KO adipose tissue macrophages (Supplementary Fig. 8g, h). Together, the above studies indicate that loss of miR-6236 influences ATM lipid accumulation but has minimal effects on gene expression in vivo during obesity.

To complement the above studies of macrophages, we also sought to examine whether cell-intrinsic miR-6236 influences insulin-dependent adipocyte functions. First, we examined miR-6236 expression in primary and 3T3-L1 adipocytes. Adipocytes expressed low levels of miR-6236 compared to ATM-derived EVs (Supplementary Fig. 8i). Next, we examined the effects of miR-6236 on adipocyte differentiation and function by differentiating preadipocytes from WT and KO mice into adipocytes ex vivo and measuring insulin-specific functions. We observed that adipocyte differentiation, PTEN expression, AKT phosphorylation, insulin-stimulated glucose uptake, and insulin-stimulated suppression of lipolysis were similar between ex vivo differentiated adipocytes from WT or KO mice (Supplementary Fig. 8j–n). The above observations indicate that miR-6236 is expressed

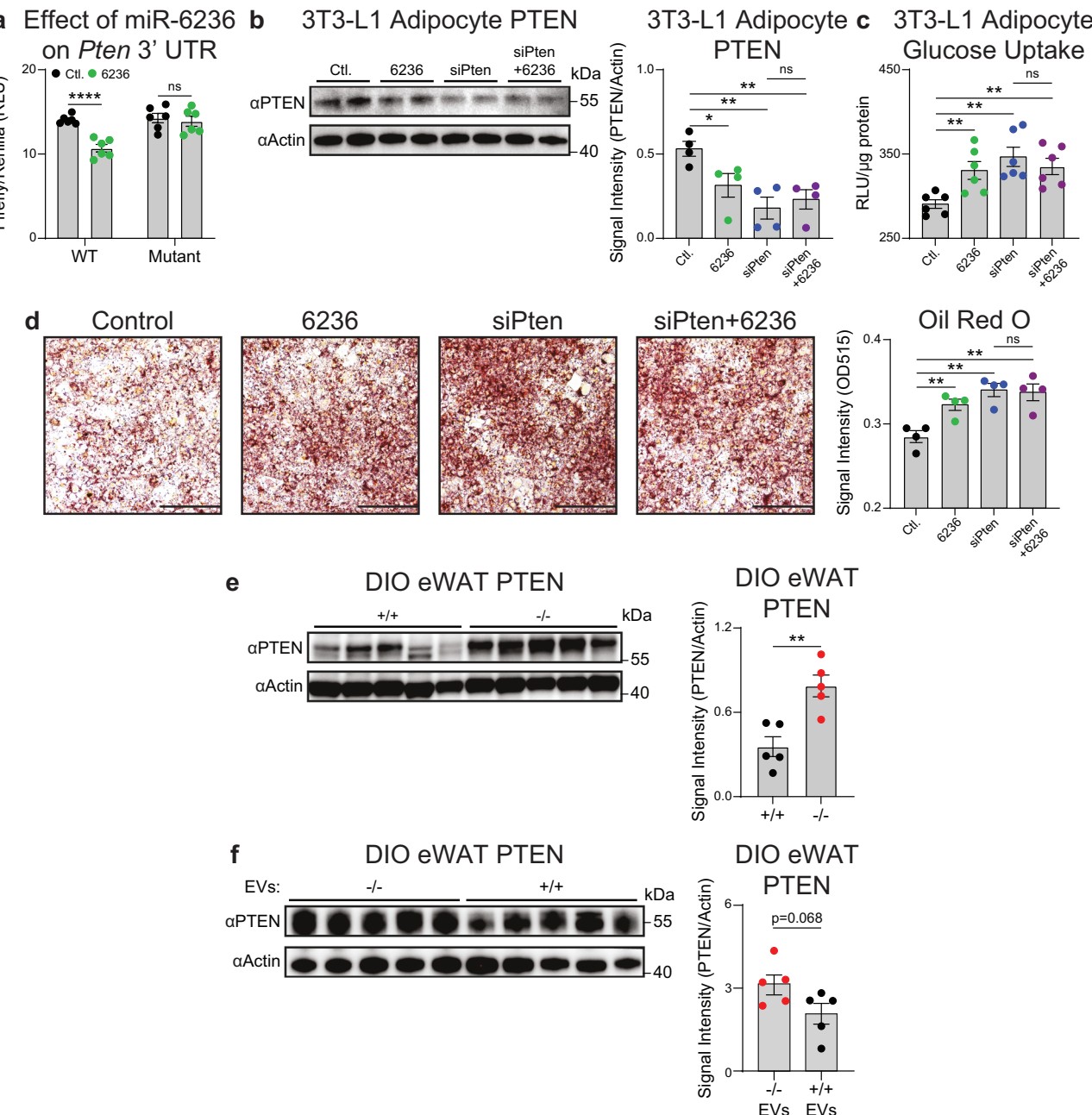

**Fig. 4 | miR-6236 promotes adipocyte insulin signaling by inhibiting *Pten*.**
**a** Dual luciferase reporter assay of scrambled control siRNA or miR-6236 binding to wild-type or miR-6236 binding sites-deleted (Mutant) *Pten* 3′ UTR cloned in pmir-GLO plasmid downstream of firefly luciferase (*n* = 6; WT, *p* < 0.0001). **b** PTEN Western blot (*n* = 2) and quantification (*n* = 4, normalized data from two independent experiments) in 3T3-L1 cells (Ctl. vs. 6236, *p* = 0.040; Ctl. vs. siPten, *p* = 0.0042; Ctl. vs. siPten + 6236, *p* = 0.0061). **c** Insulin-stimulated glucose uptake in differentiated 3T3-L1 cells (*n* = 6) (Ctl. vs. 6236, *p* = 0.0070; Ctl. vs. siPten, *p* = 0.0012; Ctl. vs. siPten + 6236, *p* = 0.0022). **d** Oil Red O-stained 3T3-L1 cells and quantification

(*n* = 4; scale bar 200 μm; Ctl. vs. 6236, *p* = 0.0082; Ctl. vs. siPten, *p* = 0.0019; Ctl. vs. siPten + 6236, *p* = 0.0050). **e** PTEN Western blot and quantification in eWAT of DIO wild-type (+/+) and knockout (−/−) male mice (*n* = 5 mice/group; *p* = 0.0033). **f** PTEN Western blot and quantification in wild-type DIO male mice injected i.p. with EVs from knockout or wild-type metabolically-activated BMDMs (*n* = 5 mice/group). Ctl. scrambled control siRNA, 6236 miR-6236 mimic, WT wild-type, siPten Pten siRNA, i.p. intraperitoneally. Data representative of ≥2 experiments. Data presented as mean ± SEM; ns not significant, *p* < 0.05, **p* < 0.01, ****p* < 0.0001, two-tailed Student's *t* test.

at a low level in adipocytes, and that cell-intrinsic miR-6236 does not influence the development or tested insulin-dependent functions of adipocytes.

To formally establish whether myeloid-derived miR-6236 contributes to the phenotype observed in KO mice, we generated miR-6236 conditional knockout mice (*miR-6236^{fl/fl}* mice). To do so, LoxP sequences were inserted flanking the endogenous *miR-6236* locus (Supplementary Fig. 9a). DNA gel genotyping (Supplementary Fig. 9b)

and in vitro treatment of BMDMs with TAT-CRE recombinase (Supplementary Fig. 9c) were consistent with the functional insertion of flanking LoxP sequences. *miR-6236^{fl/fl}* mice were bred to mice expressing Cre recombinase under the control of the endogenous *Lyz2* promoter (LysMCre) to generate mice in which immune cells of the myeloid lineage were deficient in miR-6236 (MKO). miR-6236 was absent from BMDM EVs derived from MKO mice (Supplementary Fig. 9d), establishing the functionality of this mouse model.

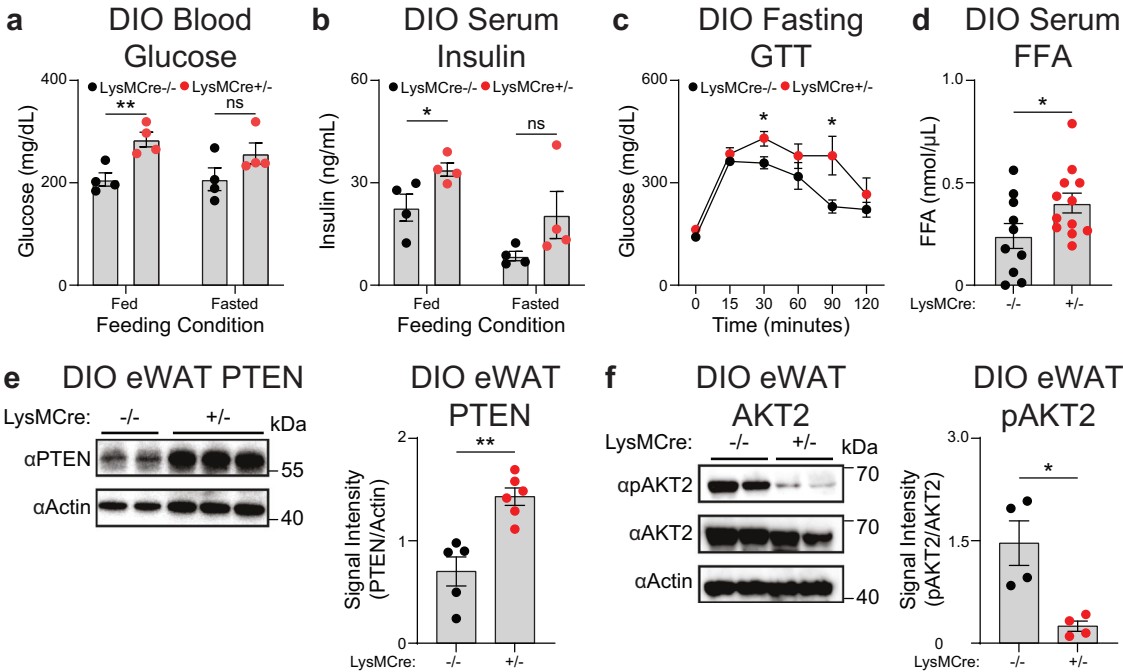

**Fig. 5 | Myeloid-specific deletion of miR-6236 exacerbates obesity-associated metabolic outcomes. a** Fed and fasted (5 h) blood glucose level in DIO *miR-6236*^fl/fl wild-type (LysMCre−/−) and myeloid-specific knockout (LysMCre +/−) male mice (*n* = 4 mice/group; Fed, *p* = 0.0070). **b** Fed and fasted (5 h) serum insulin level in DIO *miR-6236*^fl/fl male mice (*n* = 4; *p* = 0.045). **c** Intraperitoneal glucose tolerance test in DIO *miR-6236*^fl/fl male mice (*n* = 4 mice/group; 30 min., *p* = 0.034; 90 min., *p* = 0.048). **d** Serum free fatty acid (FFA) level in DIO *miR-6236*^fl/fl LysMCre−/− (*n* = 10) and LysMCre +/− (*n* = 12) male mice (*p* = 0.049). **e** PTEN Western blot (one representative experiment) and quantification (normalized data from two independent experiments) in eWAT of DIO *miR-6236*^fl/fl LysMCre−/− (*n* = 2 for Western blot, *n* = 5 for quantification) and LysMCre +/− (*n* = 3 for Western blot, *n* = 6 for quantification) male mice (*p* = 0.0013). **f** AKT2 and pAKT2 Western blot (*n* = 2) and quantification (*n* = 4, normalized data from two independent experiments) in eWAT of DIO *miR-6236*^fl/fl male mice (*p* = 0.011). Data representative of ≥2 experiments. Data presented as mean ± SEM; ns not significant, *\*p* < 0.05, *\*\*p* < 0.01, two-tailed Student's *t* test.

Like whole-body KO mice, myeloid-specific deficiency of miR-6236 did not influence DIO-induced weight gain in either male or female mice (Supplementary Fig. 9e, f). However, both male and female DIO MKO mice displayed elevated blood glucose and serum insulin levels in the fed state as compared to DIO sex-matched Cre-negative littermate controls (Fig. 5a, b and Supplementary Fig. 9g, h). DIO MKO mice also had worse fasting glucose intolerance and elevated serum free fatty acids as compared to DIO Cre-negative littermate controls (Fig. 5c, d). Finally, eWAT of DIO MKO mice had a higher level of PTEN (Fig. 5e), and reduced levels of phosphorylated AKT2 (Fig. 5f), compared with DIO Cre-negative littermate controls. Together, these data indicate that myeloid cell-derived miR-6236 influences adipose tissue insulin signaling and global metabolic homeostasis during obesity.

## A human miR-6236 homolog is inversely correlated with obesity-associated outcomes

We next sought to determine if a miR-6236 homolog is present in the human genome. Discovering a novel miRNA gene solely based on mature sequence alignment or small RNA-seq datasets is unreliable as short RNA reads can match nonspecifically to multiple genomic locations. Further, reads generated from degraded RNA fragments may be mistaken as true mature miRNAs. Since miRNAs are first transcribed in the form of much longer primary miRNA transcripts (pri-miRNAs) before being cleaved into pre-miRNAs by Drosha and mature miRNAs by Dicer, we first attempted to identify a homologous pri-miRNA transcript. As pri-miRNAs are transient and rapidly processed by Drosha, they are difficult to detect by conventional RNA-seq in normal cells. To overcome this limitation, we assembled human RNA transcripts from a published RNA-seq dataset generated from conditional Drosha knockout cells that are enriched for pri-miRNA transcripts[25].

We discovered a novel, 135 nucleotide-long transcript that maps to an unannotated chromosome location (GRCH38 Chr4: 69431035-69430901, '-' strand), hereafter, referred to as *pri-hsa-MIR-6236*. The *pre-hsa-MIR-6236* and *pre-mmu-miR-6236* loci have a sequence alignment of 90 nucleotides with ~88% (79/90) sequence identity (Supplementary Fig. 10a). *pri-hsa-MIR-6236* has a predicted miRNA-like secondary structure in the aligned region that we designated as pre-hsa-miR-6236 (Genome coordinate: GRCH38 Chr4: 69430994-69430905; '-' strand) (Supplementary Fig. 10b).

We next mapped small RNA-seq and Argonaute HITS-CLIP reads from published studies to *pre-hsa-MIR-6236*[26–28]. We detected abundant mature miRNA fragments that mapped to the *pre-hsa-MIR-6236* transcript (Fig. 6a). Analysis of a previously published adipocyte ATAC-seq dataset revealed the presence of open chromatin along the *pre-hsa-MIR-6236* locus (Fig. 6a)[29], further demonstrating a feature of a transcriptionally active gene. Finally, we analyzed previously published human serum and subcutaneous adipose tissue small RNA-seq datasets from lean and obese male subjects to determine whether obesity influences hsa-MIR-6236 expression[27]. We found that hsa-miR-6236 expression was elevated in obese subjects in both tissue types (Fig. 6b, c). In obese subjects, the level of hsa-MIR-6236 in serum was elevated to the extent that it is the sixth most abundant miRNA detected (Fig. 6d). Consistent with our findings in mice, hsa-miR-6236 inhibited translation of human PTEN by directly binding to the 3'UTR of its mRNA (Supplementary Fig. 10c). Furthermore, transfection of human primary adipocytes with hsa-MIR-6236 led to reduced PTEN protein levels and increased insulin-stimulated glucose uptake in vitro (Supplementary Fig. 10d, e). Together, these data indicate that the human genome harbors a miR-6236 homolog that shares molecular characteristics and functions with its murine counterpart.

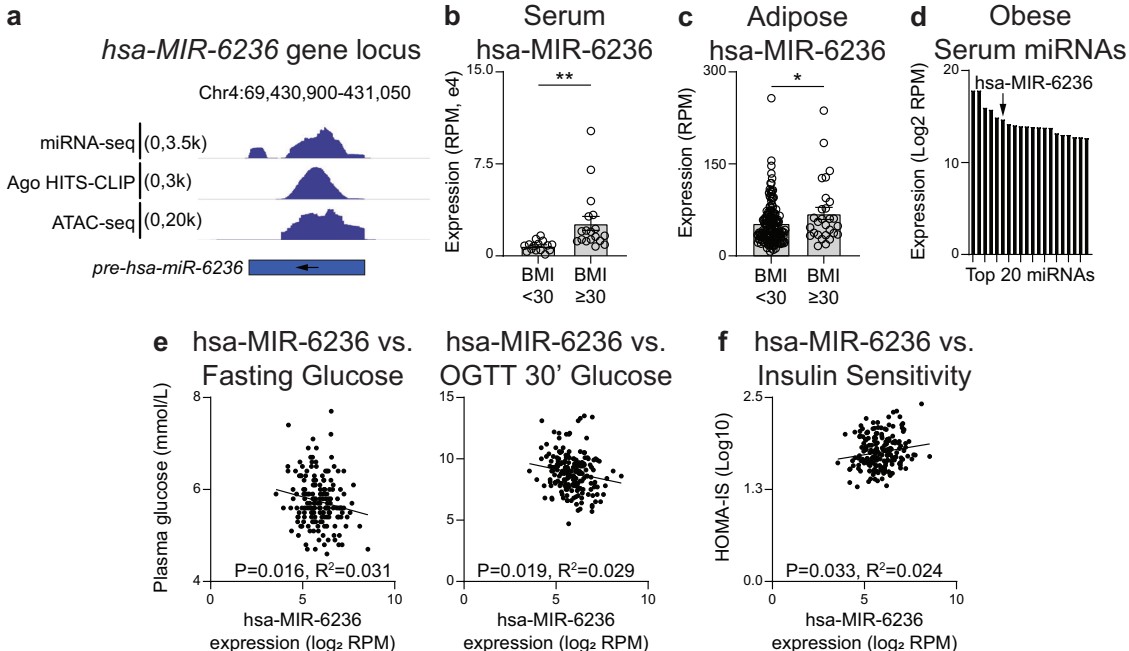

**Fig. 6 | Human MIR-6236 is elevated during obesity and correlates with blood glucose level and insulin sensitivity. a** Human (hsa) MIR-6236 genomic locus (GRCH38, Chr4: 69430994-69430905; '-' strand) and alignment of adipose tissue or adipocyte sequencing reads. **b** Expression level of hsa-MIR-6236 in serum of lean ($n = 15$) and obese ($n = 18$) male subjects measured by small RNA sequencing ($p = 0.0066$). **c** Expression level of hsa-MIR-6236 in subcutaneous adipose tissue of lean ($n = 161$) and obese ($n = 28$) male subjects measured by small RNA sequencing ($p = 0.016$). **d** The 20 most abundant miRNAs in the serum of obese male subjects given in (**b**) with hsa-MIR-6236 indicated. **e** Correlation between hsa-MIR-6236 expression level in subcutaneous adipose tissue and fasting blood glucose level or OGTT 30-min blood glucose ($n = 189$). **f** Correlation between hsa-MIR-6236 expression level in subcutaneous adipose tissue and patient HOMA-IS score ($n = 189$). Ago Argonaute, HITS-CLIP high throughput sequencing-crosslinking immunoprecipitation, ATAC assay for transposase-accessible chromatin, OGTT oral glucose tolerance test, HOMA-IS homeostatic model assessment-insulin sensitivity. Data presented as mean ± SEM; *$p < 0.05$, **$p < 0.01$, two-tailed Student's $t$ test for **b**, **c**; simple linear regression with two-sided Pearson correlation for **e**, **f**.

Finally, we sought to determine whether hsa-MIR-6236 is associated with any obesity-associated outcomes. To do so, we analyzed adipose tissue small RNA-seq data from the Metabolic Syndrome in Men (METSIM) cohort[30]. METSIM is a large population-based study consisting of 10,197 Finnish male participants examined between 2005–2010 for numerous cardiometabolic traits such as BMI, oral glucose tolerance, insulin sensitivity, and various serum metabolites. We measured the expression level of hsa-miR-6236 in subcutaneous adipose tissue of 200 METSIM study participants by reanalyzing published small RNA-seq data[27]. Then, we performed Pearson correlations between hsa-miR-6236 expression level and cardiometabolic traits. Notably, hsa-MIR-6236 displayed a significant negative correlation with fasting blood glucose level ($p \leq 0.02$) and OGTT 30' blood glucose level ($p \leq 0.02$), and a positive correlation with insulin sensitivity ($p \leq 0.03$) (Fig. 6e, f). hsa-MIR-6236 expression did not correlate with other traits analyzed including subject age and cardiometabolic outcomes (Supplementary Fig. 10f). Together, these data indicate that hsa-MIR-6236 is associated with key obesity-associated outcomes and support conserved functions for murine and human miR-6236 in mammalian metabolism.

## Discussion

The discovery of a molecule with the potential to counteract insulin resistance and obesity-associated hyperglycemia is of high relevance to the development of novel therapeutic agents for T2D. Here, we identify and define the molecular and cellular action of miR-6236, a novel ATM-derived miRNA that influences adipocyte insulin signaling and global glucose homeostasis during obesity. The presence of a human homolog of miR-6236 with metabolic associations consistent with its mechanism in mice suggests that this molecule has high translational potential.

In recent years, EV-mediated delivery of miRNAs has emerged as an important mechanism by which ATMs can modulate adipose tissue insulin signaling[11,12,31]. As ATMs are a heterogeneous cell group with both beneficial and harmful effects on mammalian metabolism depending on context[7,9,32], it is critical to investigate EV miRNA cargo from isolated ATM subsets. LAMs, which have predominantly beneficial functions in the context of obesity[9], serve as an ideal cell type to discover metabolically beneficial miRNAs. In this study, we identify miR-6236 as the most abundant miRNA secreted by LAMs and show that miR-6236 counteracts the development of obesity-associated insulin resistance by inhibiting the translation of adipocyte PTEN, a well described negative regulator of the insulin signaling pathway[33].

We utilized extensive legacy and novel datasets to characterize miR-6236's expression pattern and regulation. These data indicate that, to the extent of our characterization, miR-6236 is primarily a myeloid-secreted miRNA that is enriched in adipose tissue and positively regulated by obesity and metabolic macrophage activation. While multiple myeloid cell-types express miR-6236, ATMs and LAMs are likely the predominant source in adipose tissue based on the cellular expression, EV loading, and EV secretion patterns. We also uncovered features of miR-6236 transcriptional regulation. Our data indicates that miR-6236 is transcriptionally induced by fatty acids, in particular palmitate and arachidonate, which is consistent with its expression in metabolically activated macrophages in vivo. Further, PPARγ binds to the miR-6236 locus and a PPARγ agonist (rosiglitazone) upregulates miR-6236 expression. These observations are consistent with known features of transcriptional regulation in metabolically activated macrophages[34–36], and may indicate a potential mechanism by which thiazolidinediones exert their beneficial metabolic effects in the context of obesity-associated metabolic dysfunction.

Our data indicate that miR-6236 can act in a macrophage-extrinsic manner to regulate adipocyte insulin-dependent functions in eWAT. The fact that LAMs secrete more miR-6236-containing EVs than other adipose tissue myeloid cells and are more predominant in eWAT compared to other adipose tissue depots, may explain why we detected a trend towards reduced glucose uptake in eWAT but no other adipose tissue depots of DIO KO mice. In addition, we also observed a trend towards reduced glucose uptake by the vastus lateralis muscle of DIO KO mice in our study. As such, there could be both paracrine and endocrine-like functions for miR-6236-containing EVs. Nevertheless, as our clamp studies were likely underpowered, future experiments will be required to better establish tissue-specific functions of miR-6236.

In addition, we observed minimal macrophage-intrinsic effects of miR-6236 in vitro. However, we did observe significant changes in LAM lipid accumulation and minimal changes in ATM and LAM gene expression in vivo when comparing DIO KO to DIO WT mice. The current study is not able to determine whether these observed changes are due to primary loss of miR-6236 or secondary effects of adipose tissue dysfunction in miR-6236-deficient mice, therefore, we cannot rule out macrophage-intrinsic functions for miR-6236. It is also notable that miR-6236 is expressed by non-ATM myeloid cells (e.g., neutrophils, monocytes, and Kupffer cells), and that it could be expressed by other, non-myeloid cells as well. These other sources of miR-6236 may be physiologically relevant both in obesity and in other disease settings. Conversely, miR-6236 seems to be expressed at a very low level by adipocytes, which is perhaps the reason why adipocytes deficient in miR-6236 show similar differentiation and insulin-dependent functional capacity as miR-6236-sufficient adipocytes. Nevertheless, there could be adipocyte-intrinsic functions for miR-6236 that we have not detected in our study.

Our data indicate that miR-6236 acts as a regulator of adipose tissue insulin signaling and systemic glucose homeostasis during obesity. Herein, we characterize the effects of miR-6236 on PTEN in detail. However, we also show that miR-6236 can bind to and regulate the expression of *Prkca* mRNA. The regulation of multiple molecules that are central to insulin signaling may explain why miR-6236 KO mice have such a profound dysregulation of systemic glucose homeostasis. As an example, the DIO model in C57BL/6 J mice often leads to weight gain and hyperinsulinemia, with limited hyperglycemia[37]. In contrast, global deletion of miR-6236 results in marked obesity-associated hyperglycemia in the diabetic range (>250 mg/dl), without detectable difference in weight gain. It is also notable that female miR-6236 knockout mice develop obesity-associated hyperglycemia and hyperinsulinemia, despite the resistance of female mice to DIO-related metabolic complications[38]. In addition, KO mice display greater adipose tissue lipolysis, and reduced AKT2 phosphorylation, despite having an elevated serum insulin level, indicating a significant disruption in insulin-dependent adipocyte functions. While myeloid-specific deletion of miR-6236 recapitulated all the phenotypes of whole-body KO mice that we tested, the magnitude of the effect was slightly less for some measures. This discrepancy suggests that there may be non-myeloid sources of miR-6236 that are physiologically relevant in the context of obesity.

MicroRNA sequences and their binding sites in the 3′UTR of target genes are highly conserved across species[39,40]. However, ~50% of mouse miRNAs do not have a known human homolog which limits their translational potential. As an example, the current miRbase release lists 1917 miRNAs in the human genome and 1234 miRNAs in the mouse genome, respectively. Of these, only 622 miRNAs are homologs between the two species[41]. In addition, discovery of miRNA genes is challenging due to their smaller size and complex biogenesis[42]. To overcome these challenges, we performed extensive re-analyses of legacy transcriptomic and epigenomic sequencing data coupled with homolog sequence alignment and RNA secondary structure prediction for human MIR-6236 discovery and annotation. The newly annotated MIR-6236 meets all the characteristics of a miRNA gene: the existence of all 3 miRNA transcript forms in the cell (pri-mRNA, pre-miRNA, and mature sequences), miRNA-like stem-loop structure in the pri/pre-miRNA transcripts, sequence conservation, and euchromatic nature of the predicted gene locus. In addition to sequence and functional conservation between mice and humans, the presence of abundant MIR-6236 in serum of obese patients, along with the high degree of association that MIR-6236 has with fasting glycemia, indicate that MIR-6236 secretion by ATMs may be a conserved mechanism to provide metabolic adaptation during periods of excessive caloric intake.

miRNA-based therapeutics are an exciting new field that is growing rapidly[43]. In addition, circulating miRNAs have been proposed as disease biomarkers, relying on the fact that tissues enhance or suppress the release of particular miRNAs depending on the context[44–49]. Here, we show that miR-6236-containing EVs are sufficient to modulate insulin singling in adipocytes. However, MIR-6236 could be delivered to adipocytes by other means such as via cell-free miRNA delivery mechanisms. Future studies of this molecule are necessary to determine if MIR-6236 is sufficient to influence obesity-associated insulin resistance and hyperglycemia as a therapeutic agent. Similarly, future work is required to better understand the extent to which detection of circulating MIR-6236 is a useful biomarker for diagnosis and/or monitoring of T2DM.

## Methods
(see Supplementary Data 1 for details of reagents)

### Ethics declaration
All protocol involving research animals were approved by The Children's Hospitals of Philadelphia Institutional Animal Care and Use committee (IACUC # 19-001324).

### Mouse lines
miR-6236 KO and LoxP mouse lines were generated on a C57BL/6 J (B6) background by the University of Pennsylvania CRISPR-Cas9 Mouse Targeting Core. TREM2 KO mice were purchased from Jackson Labs (Strain #027197). Genomic modifications were confirmed by Sanger sequencing. All experiments were performed on experimental mice or appropriate age and sex matched littermate or B6 wild-type controls. miR-6236 KO and fl/fl mice are available from the Hill Lab upon reasonable request.

### Husbandry, dietary treatment, and euthanasia of mice
Mice were housed in five animal cages under specific pathogen free conditions with standard bedding and automated running water supply under a 12 h light/dark cycle. Mice were gently restrained by the scruff for a brief period to minimize stress or discomfort during and allow for movement between cages or the administration of injections. All mice were fed a normal chow diet (LabDiet 5015) until the age of 6 weeks, after which obesity was induced in DIO mice by feeding a high fat diet (Research Diets D12492, 60% kcal fat) *ad libitum* from 6 to 18–24 weeks[8]. Where indicated, mice were maintained on a control fat diet (CFD, Research Diets D12450B, 10% kcal fat) from 6–18 weeks. Male mice were preferred due to known susceptibility to diet-induce obesity and sequela, though key observations were tested in female mice. Mice were euthanized via carbon dioxide ($CO_2$) exposure administered using a low-flow $CO_2$ regulator at an approximate rate of 30–70% of the chamber volume per minute. $CO_2$ flow was maintained for at least 1 min after apparent clinical death.

### Fed and fasting blood glucose measurement
Blood was collected from the tail vein. Glucose was measured using Blood Glucose Strips (Clarity Diagnostics) under fed and brief fasting (~5 h) conditions.

## Intraperitoneal glucose tolerance test

Mice were fasted overnight (-12–16 h) in a clean cage with Alpha Dry bedding. On the day of the experiment, mice were weighed and given an intraperitoneal injection of glucose solution at a dose of 2 grams of glucose/kg body weight. Blood glucose levels were measured before (0 min) and after (15, 30, 60, 90, 120 min) glucose administration as above. Maximum values were recorded as 610 mg/dL.

## Collection of serum and tissues

Mice were euthanized by $CO_2$ asphyxiation. Serum was isolated by allowing blood samples to coagulate at room temperature for ~30 min followed by centrifugation ($1000 \times g$ for 10 min) and storage at −80 °C until future use. Major metabolic organs and tissues were dissected, weighed, and immediately used or flash frozen in liquid nitrogen and stored at −80 °C until future use.

## Measurement of metabolites in serum and tissue

Tissue lysates were prepared from frozen samples in suitable lysis buffers depending on the metabolite to be analyzed and kit to be used. Tissue was homogenized using the OMNI Bead-Ruptor Elite homogenizer (OMNI International). The following kits were used to measure their respective metabolite in serum or tissue lysate following the manufacturer's instructions: Free Fatty Acid Assay Kit (Abcam), Free Glycerol Assay Kit (Abcam), Triglyceride Assay Kit (Abcam), and Cholesterol Assay Kit – HDL and LDL/VLDL (Abcam). Metabolite levels were normalized to the protein content of the lysate.

## Ex vivo glucose uptake

eWAT was cut into 1–2 mm pieces and placed into digestion media (DMEM with 1% BSA, 6.2 mg/ml Collagenase Type I, and 2.4 mg/ml Dispase II). Tissue was homogenized using a GentleMACS (Miltenyi Biotech) and adipocytes were separated from stromal vascular fraction (SVF) by centrifugation. Purified adipocytes were resuspended in DMEM with 10% FBS and 3 μg/ml insulin and incubated for ~30 min. Cells were centrifuged, and the adipocyte layer was collected into a cell culture plate. Glucose uptake was performed on insulin-stimulated adipocytes using Glucose Uptake-Glo™ Assay kit (Promega).

## Ex vivo lipolysis assay

Ex vivo lipolysis assay was performed as described previously[50], with some modifications. Briefly, eWAT was excised, weighed, and placed into DMEM with 2% BSA. The tissue was cut into ~5–7 equal pieces and distributed into fresh 1 ml DMEM with 2% BSA for preincubation (~20 min). Tissue pieces were subsequently transferred to fresh 1 ml DMEM with 2% BSA and 3 μg/ml insulin. After 20 min, tissue pieces were transferred to fresh 1 ml DMEM media with 2% BSA and 3 μg/ml insulin for 60 min and the assay media was collected. Glycerol content in the media was measured using Free Glycerol Assay Kit (Abcam) and normalized to tissue weight.

## Insulin and Glucagon ELISA

Serum insulin and glucagon levels were measured in 2 μl of serum sample using Ultra-Sensitive Mouse Insulin ELISA kit (Crystal chem) and Mouse Glucagon ELISA Kit (Crystal chem), respectively.

## Flow cytometry and cell sorting

Blood was collected from euthanized mice via eye bleeding into blood collection tubes with 10.8 mg EDTA (Fisher Scientific). Blood was treated with RBC Lysis Buffer (Tonbo Biosciences) and filtered through a 70 μM filter. Spleen, femurs, and eWAT depots were dissected from euthanized mice. Spleen samples were passed through a 70 μM filter. Femurs from euthanized mice were dissected and flushed with cold PBS and passed through a 70 μM filter. eWAT samples were finely minced and digested in DMEM with 1 mg/mL Collagenase type IV and 10 mg/mL DNase I for 30 min at 37 °C in a shaking incubator. Cells were filtered through a 100 μM filter and adipocyte fraction was removed following centrifugation. All samples were treated with RBC Lysis Buffer (Tonbo Biosciences), washed in MACS buffer (1 x PBS, 2% fetal bovine serum, 2 mM EDTA, 25 mM HEPES), and Fc-receptor blocked (except bone marrow samples) with CD16/32 (2.4G2, 1:50, BD) prior to staining for flow cytometry. Cells were stained with anti-mouse fluorochrome-conjugated monoclonal antibodies, quantified using an Aurora flow cytometer (Cytek), and analyzed using FlowJo (BD).

All antibodies were used at 1:500 dilution unless noted below. For eWAT samples the following antibodies were used: CD45.2- BUV395, Live/Dead Blue (1:2400), CD9- BV421, F4/80- BV605, Ly6g- BV650, CD11b- BV785, BODIPY 493/503 (1:2400), CD64- PE, I-A/I-E- PE/Dazzle 594, CD11c- PE-Cy5.5, Ly6c- APC, and SiglecF- APC-Cy7. For blood and spleen samples the following antibodies were used Live/Dead Blue (1:2400), CD45- BUV 395, CD11c- BUV737, CD117- BV421, I-A/I-E- Pacific Blue, Ly6c- BV510, F4/80- BV605, Ly6g- BV650, CD64- BV711, CD11b-BV785, Fcer1a- PerCP-Cy5.5, CD19- PE CF594, CD8a, Ter119, NK1.1-PE-Cy5, CD4- APC, CD49b- AF700, and Siglec F- APC-Cy7. For bone marrow samples the following antibodies were used: Live Dead Blue (1:2400), CD45- BUV 395, CD48- BUV 563 (1:100), CD41- BUV661 (1:100), CD117- BV421 (1:100), I-A/I-E- Pacific Blue (1:100), Ly6c- BV510, CD150- BV605 (1:100), Ly6g- BV650, CD16/32- BV711, CD11b- BV785, CD34- FITC (1:100), Fcer1a- PerCP-Cy5.5 (1:100), CD135- PerCP-Fluor710 (1:100), CD125- PE (1:100), Sca-1- PE-Dazzle (1:100), CD3, NK1.1, B220, Ter 119, IL-7R- PE-Cy5, Integrin b7- AF647, and CD115- APC-Cy7 (1:100).

ATMs (CD11b+, F4/80+, CD64+, CD9−, CD63−), LAMs (CD11b+, F4/80+, CD64+, CD9+, CD63+), monocytes (CD11b+, Ly6c+), and neutrophils (CD11b+, Ly6g+) were identified in eWAT of obese B6 mice as previously described[8], and sort-purified using an FACSAria (BD) or Aurora CS sorter (Cytek).

## Differentiation of 3T3-L1 cells and primary preadipocytes

3T3-L1 cells (ATCC CL-173) were grown in tissue culture treated plates until cells reached 100% confluency, and then incubated an additional 48 h. Adipocyte differentiation was induced by induction media consisting of DMEM with 10% FBS, 500 μM IBMX, 1 μM dexamethasone, and 3 μg/ml insulin. After 2 days, induction media was replaced with adipocyte maintenance media consisting of DMEM with 10% FBS and 3 ug/ml insulin. Cells were grown in maintenance media until fully differentiated into lipid laden adipocytes (-10–15 days).

eWAT from mice or human subcutaneous adipose tissue (obtained in a de-identified manner from the Penn Human Metabolic Tissue Bank) were dissected, digested, and processed, as described above. The SVF pellet was resuspended in media, filtered through a 40 μM cell strainer, and centrifuged. The SVF pellet was resuspended in preadipocyte expansion media consisting of DMEM/F12 with 10% FBS and 10 ng/ml FGF2 and directly plated into a 24-well plate. Cells were incubated for an additional 48 h after reaching 100% confluency and then differentiation was induced using media consisting of DMEM/F12 with 10% FBS, 500 μM IBMX, 1 μM rosiglitazone, 1 μM dexamethasone, and 3 μg/ml insulin. Two days post-induction of differentiation, cells were grown in adipocyte maintenance media (DMEM/F12 with 10% FBS, 1 μM rosiglitazone, and 3 μg/ml insulin) until fully differentiated (7-10 days).

## Ex vivo bone marrow-derived macrophage (BMDM) differentiation, activation, and proliferation assay

Femurs from euthanized mice were dissected and flushed with cold PBS to collect bone marrow cells. Cells were filtered through a 40 μM cell strainer and centrifuged. Cells were resuspended in BMDM differentiation media consisting of DMEM with 10 ng/ml M-CSF and plated into 10 cm plates and cultured until fully differentiated (6–7 days). Differentiated cells were stripped from a 10 cm plate and centrifuged. Cell pellets were resuspended in M1 polarization media (DMEM with 10 ng/ml M-CSF, 5 ng/ml LPS and 15 ng/ml IFN-γ), M2 polarization

media (DMEM with 10 ng/ml M-CSF and 15 ng/ml IL-4), or media (DMEM with 10 ng/ml M-CSF) supplemented with 300 μM of palmitate or arachidonate and plated into 12-well assay plates. In addition, in experiments using rosiglitazone, BMDMs were treated with 1 μM Rosiglitazone or equal volume of DMSO in addition to media (DMEM with 10 ng/ml M-CSF containing either arachidonate or BSA control). RNA for qPCR was extracted from cells after 24 h of polarization.

To quantify macrophage proliferation, 525,000 bone marrow cells were plated in a 24-well plate and live cells were quantified after 5 days of incubation in BMDM differentiation media.

### Purification of extracellular vesicles (EVs) for transcriptomics and treatments

For miRNA sequencing, LAMs were cultured overnight in 1 mL of serum-free media followed by isolation of the cellular fraction by ultracentrifugation and isolation of EVs from culture supernatant using ultracentrifugation ($100,000 \times g$ for 2 h, twice on the Optima L-90K Ultracentrifuge using a Beckman Coulter Ti-45 fixed angle rotor). Small RNAs were isolated and RNA-seq libraries generated using the Nucleospin miRNA isolation kit (Macherey-Nagel) and the SMARTer small RNA-seq kit (Clontech) following the manufacturer's instructions.

For in vitro treatments, BMDMs from WT or KO mice were differentiated and metabolically-activated with palmitate, as described above. Cells were grown in metabolic activation media with EV-depleted FBS (Neuromics) for 48 h, and then EVs were purified from the cell culture media using total EV isolation kit (Invitrogen, 4478359).

For in vivo treatments, BMDMs from WT or KO mice were differentiated and metabolically-activated with palmitate, as described above. Cells were grown in metabolic activation media with EV-depleted FBS (Neuromics) for 48 h, and then EVs were purified from the cell culture media by ultracentrifugation ($100,000 \times g$ for 2 h, twice on the Optima L-90K Ultracentrifuge using a Beckman Coulter Ti-45 fixed angle rotor).

### EV administration in vivo and in vitro co-culture with primary adipocytes

For in vivo treatment, isolated EVs were quantified by measuring the total EV protein content by BCA. Then, DIO male mice were given two doses of EVs 3 days apart by intraperitoneal injection (40 μg protein equivalent of EVs/mice/dose). Two days after the second dose, blood glucose and insulin level were measured, and mice were euthanized to harvest tissues for molecular assays.

For ex vivo co-culture, isolated EVs were resuspended in PBS (1 ml PBS/10 ml culture media). Preadipocytes were isolated from stromal vascular fraction (SVF) of WT mice and differentiated into mature adipocytes in a 24-well plates, as described above. After full differentiation of adipocytes, old culture media was replaced with a new culture media supplemented with WT or KO EVs (420 μl of media + 80 μl EVs) and incubated for 24 h. Western blot, glucose uptake, and lipolysis assays were performed after 24 h of incubation.

### TAT-Cre treatment of BMDM for functional validation of LoxP sites in *miR-6236*^*fl/fl*^ mice

BMDMs were isolated and ex vivo differentiated, as described above. After full differentiation, cells were stripped and plated in a 24-well plates and incubated for 24 h. Cells were treated with either TAT-Cre (5 μM, Millipore Sigma) or BSA and incubated an additional 24 h. Genomic DNA was extracted from cells and fragment size was analyzed after PCR amplification.

### In vitro glucose uptake assay

3T3-L1 cells were differentiated, as described above, and transfected with control siRNA, 30 nM miR-6236 mimic, 30 nM *Pten* siRNA, or 30 nM each of miR-6236 mimic and *Pten* siRNA using Lipofectamine

RNAiMAX transfection reagent (Invitrogen). Two days after transfection, culture media was removed and replaced with serum free media (DMEM with 2% BSA) and cell were incubated for ~3–5 h (basal glucose uptake). For insulin-stimulated glucose uptake, cells were insulin stimulated for ~30 min in DMEM media with 10% FBS and 3 μg/ml insulin. Glucose uptake assay was performed using Glucose Uptake-Glo™ Assay kit (Promega).

### In vitro lipolysis assay

3T3-L1 cells were differentiated, transfected, serum-starved, or insulin stimulated, as above. For basal lipolysis, serum free media was replaced with fresh serum free media and cells were incubated for ~3–6 h. For insulin stimulated lipolysis, insulin media in the well was replaced with fresh insulin media and incubated for ~3–6 h. After incubation, media was collected, and glycerol content was measured using Free Glycerol Assay Kit (Abcam).

### In vitro lipogenesis assay

3T3-L1 cells were differentiated, as described above. 3–4 days post-induction, just before appearance of visible lipid droplets, cells were transfected, as described above. Three days after transfection, cells were stained with Oil Red O dye for total cellular lipids and imaged using an EVOS FL Auto microscope (Thermo Scientific). Oil Red O dye from the stained cells was extracted using isopropanol and optical density (OD) was measured at 515 nm wavelength.

### Dual luciferase reporter assay for miRNA target validation

A segment of the *Pten* 3′UTR (1067 bp) harboring two miR-6236 predicted binding sites for mature sequences miR-6236e and miR-6236f was amplified from genomic DNA using primer pairs given in Supplementary Data 1. A mutant *Pten* 3′UTR with an identical sequence other than deletion of the two miR-6236 predicted binding sites was synthesized by Life Technologies Corporation. Similarly, WT and mutant 3′UTRs for human *Pten* and murine *Prkca* were synthesized by Life Technologies Corporation. 3′ UTRs and pmirGLO plasmid vector (Promega) were separately restriction digested (with Nhei and Sbfl), purified, and the 3′UTR inserts were separately ligated into the plasmid just downstream of firefly luciferase gene open reading frame using T4 DNA ligase. Recombinant plasmid vectors were independently transfected into 293 T (ATCC CRL-3216) cells along with control siRNA or miR-6236 mimic (miR-6236e or miR-6236f). The next day, firefly and renilla luciferase protein levels were measured using the Dual Luciferase Assay System (Promega) in a Biotek Synergy HTX plate reader with a dual injector (Agilent). Firefly luciferase signal was normalized to renilla luciferase signal to correct for differences in transfection efficiency among samples.

### Western blots

Tissue samples and cultured cells were homogenized in T-PER™ tissue protein extraction reagent (Thermo Scientific) with Halt™ protease inhibitor cocktail (Thermo Scientific) using a Bead Ruptor Elite homogenizer. Samples were separated on 4–12% Bis-Tris precast mini gels and protein samples were transferred to PVDF membrane using Trans-Blot® Turbo™ transfer system (Bio-Rad). Primary antibodies used were anti-phospho AKT2 (Ser474) (Cell Signaling, 8599 S; 1:5000 dilution), anti-AKT2 (Cell Signaling, 3063 S, 1:5000 dilution), anti-PTEN (Santa Cruz Biotechnology, sc-7974, 1:1000 dilution), anti-p-HSL (Cell Signaling, 4126, 1:7000 dilution), anti-HSL (Cell Signaling, 4107, 1:7000 dilution) anti-actin (Sigma Aldrich, A2066-100UL, 1:5000 dilution), and anti-CD9 (BD Bioscience, 553758, 1:1000 dilution). Secondary antibodies used were anti-mouse IgG HRP (Cell Signaling, 7076 S, 1:10,000 dilution), anti-rabbit IgG HRP (Novus Biologicals, HAF008, 1:5000 dilution), and anti-rat IgG HRP (Novus Biologics, 22109, 1:10,000). Protein bands on the membrane were detected by Enhanced chemiluminescent (ECL) substrate using a ChemiDoc

MP (BioRad). Uncropped Western Blots are provided in Source Data File.

### RNA extraction, qPCR, and RNA-seq library preparation

Total RNA was extracted from frozen tissue, cultured cells, or sort-purified cells using Trizol Reagent (Ambion, 15596018) following manufacturer's recommendation. cDNA was synthesized using the Verso cDNA synthesis kit (Thermo Scientific, AB1453B) and qPCR reaction was carried out using PowerUp SYBR green master mix (Thermo Scientific, A25741) and a QuantStudio 12 K Flex or 7 K Real-Time PCR System (Applied Biosystem). miRNA qPCR was performed using the Taqman advanced miRNA assay kit (Thermo Fisher Scientific, A25576) following the manufacturer's instructions. Actin (*Actb*) and *miR-21* were used as reference transcripts for normalization of mRNA and miRNAs, respectively. Primers used in qPCR are given in Supplementary Data 1.

For tissue-level RNA-seq of eWAT and liver, libraries were prepared using the TruSeq RNA library prep kit (Illumina, RS-122-2001) and standard Illumina protocol. RNA-seq libraries were sequenced at single or paired-end, 75- to 100-bp read length on an Illumina HiSeq 2000.

### Oil Red O staining of cultured cells

Media was removed from the wells and cells were washed with PBS (1X) followed by fixing of cells with 4% paraformaldehyde for 1 h. Cells were washed (2X) with deionized water, incubated with working solution of Oil Red O for 10 min, washed (4X) again with deionized water and then imaged. For quantification of total lipid, Oil Red O from stained cells was extracted with 100% isopropanol and OD was measured at 515 nm.

### Epididymal and inguinal white adipose tissue histology

Adipose tissue was washed in PBS and fixed in 4% buffered paraformaldehyde solution overnight at 4 °C followed by ethanol dehydration. Dehydrated samples were sent to CHOP histology core to perform hematoxylin and eosin staining. Images were acquired using EVOS FL Auto microscope (Thermo Fisher Scientific) and adipocyte size was measured using Adiposoft plugin in the Fiji application.

### Hyperinsulinemic-euglycemic clamp

Hyperinsulinemic-euglycemic clamp studies were performed by the Penn Diabetes Research Center Rodent Metabolic Phenotyping Core (University of Pennsylvania); +/− or −/− littermate mice were bred in the Hill Lab and co-housed in mixed cages to minimize variability due to environmental factors. Hyperinsulinemic-euglycemic clamp studies were performed as previously described[51,52], with some modifications. Indwelling jugular vein and carotid artery catheters were surgically implanted in the mice for infusion 7 days prior to the clamp study day. Mice were acclimated to the containers (plastic bowl with alpha dry) and fasted for 5 h prior to initiation of clamp. Jugular vein and arterial line are hooked up to the dual swivel 2 h prior to the clamp initiation. During the 90 min basal period, mice received a bolus infusion of 1.5 μCi [3-3H]-D-glucose followed by a constant infusion via the jugular vein at 0.075 μCi/min. Baseline measurements were determined in blood samples collected at −10 and 0 min (relative to the start of the clamp) for analysis of glucose and [3-3H] glucose specific activity. The clamp was started at $t = 0$ min with a primed-continuous infusion of human insulin (2.5 mU/Kg/min; Novolin Regular Insulin) and a donor blood infusion at 4.5 μL/min to prevent a 5% drop in the hematocrit. Blood glucose levels were clamped at euglycemia and tracer specific activity was maintained with a variable infusion of 50% dextrose + 3-[3H]-D-glucose (0.05 μCi/ul in D50) for 2 h. Blood samples were taken at $t = 80$–120 min for the measurement of [3-3H] glucose specific activity and glucose infusion rates were adjusted to maintain euglycemia. 120 min after initiation of clamp, 12 mCi of 2-[14 C]

deoxyglucose ([14 C]2DG) was administered as a bolus and arterial blood samples were obtained at 2, 5, 10, 15 and 25 min to determine Rg, an index of tissue-specific glucose uptake in various tissues. After the final blood sample, animals were injected with a bolus of pentobarbital, and tissues were collected and frozen in liquid nitrogen and stored at −80 °C for subsequent analysis.

Processing of samples and calculations: Radioactivity of [3-3H] glucose, [14 C]2DG and [14 C]2DG-6-phosphate were determined, as previously described[52]. The glucose turnover rate (total Ra; mg/kg/min) was calculated as the rate of tracer infusion (dpm/min) divided by the corrected plasma glucose specific activity (dpm/mg) per kg body weight of the mouse. Tissue specific glucose disposal (Rg; mmol/100 g tissue/min) was calculated, as previously described[52].

### Bioinformatic analyses

During the initial discovery of mouse miR-6236 in LAM EVs, small RNA-seq reads from the eWAT LAM EVs were aligned to the mouse genome, mm10, using STAR aligner[53]. miRNA abundance was quantified using miRBase[41] annotation and FeatureCounts[54]. Before analyzing any legacy sequencing data, barcodes and sequencing adapters in the library were removed and reads were filtered based on the quality score. All analyses, except assembly of human transcripts from Drosha KO cell lines, were performed in CLC Genomics Workbench Suite (Qiagen) using standard protocols and analysis tools. Human pri-miRNA transcripts in Drosha KO cell lines were assembled using HISAT2 sequence aligner[55] and StringTie transcript assembler[56] in Cyverse Discovery Environment Suite. For murine miRNA expression analysis, miRNA sequences in miRbase release 22.1 were used as a reference. For human miRNA expression analyses, the newly discovered hsa-MIR-6236 was also added to the miRbase reference before using it as a reference. Read tracks from small RNA-seq, Argonaute HITS-CLIP, PPARγ ChIP-seq, and ATAC-seq datasets were generated by mapping the reads to the respective human or murine genomes in CLC genomics, and read tracks were visualized by Integrated Genome Browser (IGB). Differentially-expressed genes between tissue conditions were identified as FC > 2, FDR < 0.05, >1 RPKM.

### Reporting summary

Further information on research design is available in the Nature Portfolio Reporting Summary linked to this article.

## Data availability

In-lab generated raw small RNA sequencing data from CD9 + ATM EVs fractions are deposited in Mendeley under accession number https://doi.org/10.17632/cbszxmd63x.1 (https://data.mendeley.com/datasets/cbszxmd63x/1). NCBI locations of raw public data used in this study are as follows: murine tissue and ATM derived EV small RNA-seq [(GSE119661), (GSE142677), (GSE97652)]; murine adipose tissue argonaute HITS-CLIP (GSE142677); murine Kupffer cell small RNA-seq (GSE160016); young (6 month old) and aged (24 month old) murine serum small RNA-Seq (GSE76442); murine peritoneal macrophage PPARγ ChIP-seq (GSM3022232); murine LAM ATAC-seq (GSE113583); 3T3-L1 and primary adipocyte small RNA-Seq [(PRJEB20090) (PRJEB25978)]; Human Drosha KO RNA-seq (PRJNA282167); Human serum small RNA-seq (PRJEB21747); Human adipose tissue small RNA-seq (GSE45159); Human argonaute HITS-CLIP [(GSE412720 GSE41357) and human adipocytes ATAC-seq (GSE178796). All other data required to reproduce the results described here can be found in the manuscript, figures, and supplementary material. Source data are provided with this paper.

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

## Acknowledgements

We thank the Penn Institute for Diabetes, Obesity and Metabolism sequencing core, Penn Transgenic Mouse Core, Penn Diabetes Research Center Rodent Metabolic Phenotyping Core, and CHOP Flow Cytometry Core for supporting this work. This work was funded by the National Institutes of Health (NIH; Grants K08 DK116668 and R03 DK129418 to D.A.H.; R01 DK49780 to M.A.L.), the JPB Foundation (M.A.L.), and the Children's Hospital of Philadelphia Research Institute (D.A.H.). The contents of this article represent the views of the authors and do not necessarily represent the official views of, nor an endorsement by, the funding sources. The funding sources had no role in the design or conduct of this study.

## Author contributions

B.P.: conceptualization, investigation, formal analysis, data curation, software, writing - original draft, writing - review & editing; J.C.: conceptualization, investigation, formal analysis, writing - review & editing; S.M.: investigation; N.D.: investigation; J.M.: investigation, project administration; L.D.J.: methodology; J.H.M.: methodology; P.M.T.: conceptualization; D.M.M.: conceptualization, resources; H.W.L.: formal analysis, data curation, software; M.A.L.: conceptualization, writing - review & editing, funding acquisition; D.A.H.: conceptualization, formal analysis, writing - original draft, visualization, writing - review & editing, supervision, funding acquisition. All authors approved the final manuscript as submitted and agree to be accountable for all aspects of the work.

## Competing interests

D.A.H. holds a patent on the biomedical use of miR-6236. The remaining authors have no conflicts of interest to disclose.
