## [Peer Review File · Nature Communications]

Myeloid-derived miR-6236 potentiates adipocyte insulin signaling and prevents hyperglycemia during obesityEditorial Note: This manuscript has been previously reviewed at another journal that is not operating a transparent peer review scheme. This document only contains reviewer comments and rebuttal letters for versions considered at *Nature Communications*.

REVIEWERS' COMMENTS

Reviewer #1 (Remarks to the Author):

The authors successfully replied to our comments.

Reviewer #2 (Remarks to the Author):

The group has done very elegant work on characterizing the role of miR-6236 in promoting insulin sensitivity by inhibiting neg regulators such as PTEN, and they show cell autonomous effects in vitro and in myeloid and whole body KO mice. They show using legacy data and novel datasets the clinical significance of miR-6236.

The data in GTT and ITT in mice are quite robust, but this is not support using clamp studies where all the data are non-significant. They focus on trends which are essentially non-significant and hard to reconcile with their conclusions, particularly in Fig 2 – where they show decreased glucose infusion but no changes in uptake in adipose tissue or muscle nor glucose production in liver. This is still difficult to reconcile and wonder whether more n would be needed. Some expts have n of 3-5 or 5-6 which may not be powered to detect statistical significance. To emphasize on the trend does not make a strong case to support their conclusion.

Line 196 - Together, these data indicate that miR-6236 promotes insulin sensitivity and protects

against the development of some adverse metabolic outcomes during obesity. – this is not supported by their data.